# Identifying prognostic factors for survival in intensive care unit patients with SIRS or sepsis by machine learning analysis on electronic health records

**Maximiliano Mollura**[1], **Davide Chicco**[2,3]*, **Alessia Paglialonga**[4], **Riccardo Barbieri**[1]

**1** Dipartimento di Elettronica Informazione e Bioingegneria, Politecnico di Milano, Milan, Italy, **2** Institute of Health Policy Management and Evaluation, University of Toronto, Toronto, Ontario, Canada, **3** Dipartimento di Informatica Sistemistica e Comunicazione, Università di Milano-Bicocca, Milan, Italy, **4** CNR-Istituto di Elettronica e di Ingegneria dell'Informazione e delle Telecomunicazioni (CNR-IEIIT), Milan, Italy

* davidechicco@davidechicco.it

## Abstract

### Background

Systemic inflammatory response syndrome (SIRS) and sepsis are the most common causes of in-hospital death. However, the characteristics associated with the improvement in the patient conditions during the ICU stay were not fully elucidated for each population as well as the possible differences between the two.

### Goal

The aim of this study is to highlight the differences between the prognostic clinical features for the survival of patients diagnosed with *SIRS* and those of patients diagnosed with sepsis by using a multi-variable predictive modeling approach with a reduced set of easily available measurements collected at the admission to the intensive care unit (ICU).

### Methods

Data were collected from 1,257 patients (816 non-sepsis SIRS and 441 sepsis) admitted to the ICU. We compared the performance of five machine learning models in predicting patient survival. Matthews correlation coefficient (MCC) was used to evaluate model performances and feature importance, and by applying Monte Carlo stratified Cross-Validation.

### Results

Extreme Gradient Boosting (MCC = 0.489) and Logistic Regression (MCC = 0.533) achieved the highest results for SIRS and sepsis cohorts, respectively. In order of importance, APACHE II, mean platelet volume (*MPV*), eosinophil counts (*EoC*), and C-reactive protein (*CRP*) showed higher importance for predicting sepsis patient survival, whereas, SOFA, APACHE II, platelet counts (*PLTC*), and *CRP* obtained higher importance in the *SIRS* cohort.

**Data Availability Statement:** The dataset is publically available at the following URL: https://figshare.com/articles/dataset/_C_Reactive_Protein_and_Hemogram_Parameters_for_the_

Non_Sepsis_Systemic_Inflammatory_Response_
Syndrome_and_Sepsis_What_Do_They_Mean_/
1644426.

**Funding:** The work of DC was funded by the
European Union – Next Generation EU program, in
the context of The National Recovery and
Resilience Plan, Investment Partenariato Esteso
PE8 "Consequenze e sfide dell'invecchiamento",
Project Age-It (Ageing Well in an Ageing Society)
and partially supported by Ministero dell'Università
e della Ricerca of Italy under the "Dipartimenti di
Eccellenza 2023-2027" ReGAInS grant assigned to
Dipartimento di Informatica Sistemistica e
Comunicazione at Università di Milano-Bicocca.
The funders had no role in study design, data
collection and analysis, decision to publish, or
preparation of the manuscript.

**Competing interests:** The authors have declared
that no competing interests exist.

## Conclusion

By using complete blood count parameters as predictors of ICU patient survival, machine learning models can accurately predict the survival of *SIRS* and sepsis ICU patients. Interestingly, feature importance highlights the role of *CRP* and APACHE II in both SIRS and sepsis populations. In addition, *MPV* and *EoC* are shown to be important features for the sepsis population only, whereas SOFA and *PLTC* have higher importance for *SIRS* patients.

## Author summary

Sepsis is defined as the dysregulated host response to infection causing a significant increase in patients' mortality, thus resulting in an important global health problem. Systemic inflammatory response syndrome (SIRS), an exaggerated response of the body to a noxious stressor, and sepsis, an organ dysfunction caused by a dysregulated host response to infection, are two of the most critical conditions in the intensive care unit showing high patients' mortality and resulting in an important global health problem. However, the major differences leading to an improvement in the patient conditions during SIRS and sepsis are not fully elucidated.

In this study, we assess the role of simple and easily available routinely collected blood count parameters as predictors for patients' prognosis in 1,257 patients with SIRS or sepsis. We applied and compared the performance of five distinct machine-learning models when predicting ICU patient survival. Furthermore, we investigated the feature importance of the best-performing models for each population to highlight the major differences between the two populations. Results highlight the role of C-reactive protein and APACHE II score in both populations, whereas mean platelet volume and eosinophil counts show higher importance in sepsis patients and SOFA score and platelet count show higher importance for SIRS patients.

## Introduction

Patient's outcome has long been used as primary endpoint for trials in critical care as well as for determining the patient's prognosis after treatments. Patient mortality and survival are indeed the major clinical outcomes, and they are main targets for assessing prognostic factors driving the patient conditions and the effectiveness of clinical interventions [1, 2], especially in the intensive care unit (ICU) where admitted patients are usually in very critical conditions and require constant monitoring and treatment.

In this context, sepsis represents an important global health problem accounting for about one-third of ICU deaths and its reported incidence is still increasing [3–6] and a proper and precise description of sepsis is still not available. Indeed, the definition of sepsis was subjected to several revisions during the years [7–9], and according to the Third International Consensus Definitions for Sepsis and Septic Shock [9] it is currently defined as a life-threatening organ dysfunction caused by a dysregulated host response to infection. This last update of the sepsis definition abandons the use of Systemic Inflammatory Response Syndrome (SIRS) criteria, which were recognized as presenting a lack of specificity, whereas it focuses on the life-threatening condition and the presence and progression of organ failure. However, the major

differences between SIRS and sepsis leading to a positive or negative patient outcome are still not fully elucidated in the medical literature.

In particular, Gucyetmez et al. [10] evaluated the ability of hemogram parameters, a set of medical laboratory tests providing information about the cells in a person's blood, and C-reactive protein (CRP) to distinguish non-sepsis SIRS from sepsis patients. The authors found that the combinations of CRP, lymphocytes count (LymC), and platelet count (PLTC) can be used to determine the likelihood of sepsis, however without exploring the association of these parameters with the patient survival for each population. This information can provide significant indications about the most important prognostic factors specifically for non-sepsis SIRS and sepsis patients. Also, the authors did not investigate the predictive power of the observed variables.

Especially for this last task, machine learning approaches have shown a good ability in the early identification of sepsis with data collected both from electronic health records [11–13] and from physiological vital signs monitoring [14], also providing insights about the role of each feature in a multi-variable setting. Several studies focused on predicting ICU patient outcomes focusing on mortality or survival prediction task [15–22], but multi-variable prognostic models estimating and comparing SIRS and sepsis outcomes are still lacking. The stratification of patients' risk in particular for patients undergoing infections and with sepsis is important. In fact, these patients often require prompt management and interventions, like the initiation of antibiotic therapy and the administration of fluid and vasopressors for maintaining adequate tissue perfusion and hemodynamic stability [23]. These aspects led to an increasing interest toward the prediction of the patient outcome specifically for sepsis patients, in the last few years [24–29].

In this context, simple and easily available laboratory measurements of blood cell counts (for example platelet, eosinophil, neutrophil, and lymphocyte counts) can be useful tools for patients' risk stratification.

The goal of this study is to further explore the ability of hemogram parameters in estimating the survival of ICU patients with non-sepsis SIRS and with sepsis, by applying machine learning techniques in order to estimate the survival probability of ICU patients and to investigate the role of the different parameters in a multi-variable prediction setting. Specifically, our study makes further use of the features proposed by Gucyetmez and colleagues [10] to explore the predictive power of hemogram parameters in estimating the survival probability of patients with non-sepsis SIRS and with sepsis, by comparing different machine learning approaches.

Multi-variable feature importance of the best performing models is applied to further assess the role of each feature and to highlight differences between non-sepsis SIRS and sepsis cohorts.

## Dataset

In this study, we use data retrospectively collected from 1,257 eligible medical and surgical patients admitted to the ICU's of Acibadem International Hospital and Atasehir Memorial Hospital between 1 January 2006 and 31 December 2013, Istanbul, Turkey, and made available by Gucyetmez et al. [10]. The considered cohort includes 816 (64.9%) non-sepsis SIRS and 441 (35.1%) sepsis patients.

The dataset contains the following features for each patient: Age, sex, APACHE II and SOFA scores, diagnosis (medical, elective, and emergency surgery), length of ICU stay (LOS-ICU), mortality, CRP, WBCC, NeuC, LymC, NLCR, EoC, PLTC, MPV. A detailed description of the data here used is provided in Table 1. The target variable for our analysis was survival, indicating whether the patient survived (1) or died (0) in the ICU. A quantitative

**Table 1. Description, unit of measure and range of values of each available feature in the dataset.** EC: Elective, AC: Emergency, and M: Medical. E: Male and K: Female.

| feature | description | unit of measure | values |
|---|---|---|---|
| Age | Patient's age at ICU admission | years | integer>0 |
| APACHE II | Illness severity score | ordinal | integer [0–71] |
| CRP | Acute phase reactant produced in liver | $mg/dL$ | continuous |
| Diagnosis | Reason for ICU admission | categorical | [EC, AC, M] |
| EoC | Eosinophils (cells) count | $10^3/\mu L$ | continuous |
| cohort | Indication of sepsis (1) or non-sepsis SIRS (0) | - | binary |
| Sex | Patient's sex | categorical | [E,K] |
| LOS-ICU | Patient's length of stay in ICU | days | continuous |
| LymC | Lymphocytes (cells) count | $10^3/\mu L$ | continuous |
| MPV | Mean platelet volume | $fL$ | continuous |
| NeuC | Neutrofil (cells) count | $10^3/\mu L$ | continuous |
| NLCR | Neutrophil-lymphocyte count ratio | ratio | continuous |
| SOFA | Illness severity score | ordinal | integer [0–24] |
| PLTC | Platelets count | $10^3/\mu L$ | continuous |
| WBCC | White blood cells count | $10^3/\mu L$ | continuous |
| Mortality | Patient's outcome: dead or survived | - | binary |

description of the distribution of each numeric and categorical feature for the non-sepsis SIRS (*SIRS*) and sepsis (*SEPSIS*) cohorts are reported in Tables 2 and 3. From the analysis of the target variable (*Survival*) it is possible to observe that both cohorts are unbalanced, with stronger unbalance in the *SIRS* (3.07% not survived) than in the *SEPSIS* (23.64% not survived) cohort.

## Methods

In this retrospective study, we developed predictive models of patient survival using machine learning algorithms and we evaluated the importance of features associated with patient survival using machine learning and biostatistics approaches for both the SIRS and SEPSIS populations separately (Fig 1). All the analyses were performed with the Python 3.8.3 programming language, and `scikit-learn` 1.0 and `SciPy` 1.7.1 software packages. Observations with missing information (three patients) were removed.

### Associations between features and survival

The association between the input features and patient survival was also explored with classical statistical approaches. Specifically, differences in numeric features between survived and deceased groups in each cohort were tested with the Mann-Whitney $U$ test, whereas differences in categorical features were assessed with $\chi^2$-test [30]. Statistical significance was defined for $p < 0.005$ as advocated by Benjamin et al. [31], which also accounts for multiple testing adjustments.

### Survival prediction models

We trained five machine learning models with the goal of predicting patient's survival considering the following features: *Age, Sex, SOFA, APACHE II, CRP, WBCC, NeuC, LymC, EOC, NLCR, PLTC,* and *MPV*, for both *SIRS* and *SEPSIS* cohorts. The approach consisted of 100 runs of Monte Carlo stratified Cross-Validation with 80%-20% train-test split as already proposed by Chicco et al. [32]. At each iteration, 80% of the data were used as a training set and

**Table 2. Median and interquartile range (IQR) for each numeric variable of the dataset, stratified by** *Survival* **(S: Survived, NS: Not survived, T: Total cohort), and for the** *SIRS* **and** *SEPSIS* **cohorts.**

| feature | SIRS median (IQR) | SEPSIS median (IQR) |
|---|---|---|
| APACHE II—S | 9 (5–12) | 16 (11–20) |
| APACHE II—NS | 25 (20–31) | 27 (23–31.25) |
| APACHE II—T | 9 (6–13) | 18 (14–25) |
| Age—S | 55 (36–69) | 62 (51–75) |
| Age—NS | 61 (51–74) | 65.5 (55–78.25) |
| Age—T | 55 (37–69) | 63 (51.75–76) |
| CRP—S | 2 (0.5–6.08) | 5.42 (1.38–12.66) |
| CRP—NS | 2.1 (0.5–5.66) | 6.65 (2.3–16.21) |
| CRP—T | 2 (0.5–6.07) | 5.6 (1.6–13.97) |
| EOC—S | 10 (0–40) | 10 (0–30) |
| EOC—NS | 50 (20–70) | 15 (0–40) |
| EOC—T | 10 (0–40) | 10 (0–30) |
| LOS-ICU—S | 1 (1–2) | 4 (1–9) |
| LOS-ICU—NS | 1 (1–5) | 6 (3–17.25) |
| LOS-ICU—T | 1 (1–2) | 4 (2–10) |
| LymC—S | 0.93 (0.61–1.34) | 0.71 (0.45–1.17) |
| LymC—NS | 1.22 (0.77–1.55) | 0.72 (0.42–1.12) |
| LymC—T | 0.93 (0.62–1.36) | 0.71 (0.44–1.15) |
| MPV—S | 10.1 (9.4–10.7) | 10 (9.4–10.7) |
| MPV—NS | 10 (9.1–10.4) | 10 (9.1–11) |
| MPV—T | 10.1 (9.4–10.7) | 10 (9.3–10.8) |
| NLCR—S | 10.02 (6.75–14.5) | 11.41 (7.33–17.98) |
| NLCR—NS | 8.94 (4.25–14.34) | 11.57 (6.65–21.51) |
| NLCR—T | 10.02 (6.7–14.5) | 11.48 (7.19–18.57) |
| NeuC—S | 9.22 (6.53–12.7) | 8.2 (5.5–12.7) |
| NeuC—NS | 10.17 (7.2–14.73) | 8.76 (5.7–13.24) |
| NeuC—T | 9.28 (6.56–12.73) | 8.25 (5.57–12.72) |
| PLTC—S | 191 (133–241) | 174 (105.75–256.25) |
| PLTC—NS | 172 (115–255) | 150 (82–241.5) |
| PLTC—T | 190 (132.25–241.75) | 170.5 (101–255.25) |
| SOFA—S | 1 (0–2) | 2 (0–6) |
| SOFA—NS | 9 (4–11) | 8 (7–10) |
| SOFA—T | 1 (0–2) | 4 (1–7) |
| WBCC—S | 11.23 (8.11–14.95) | 9.91 (6.87–14.38) |
| WBCC—NS | 12.14 (9.12–19.56) | 10.56 (7.38–15.5) |
| WBCC—T | 11.26 (8.17–15.05) | 10.02 (7.08–14.6) |

20% as a test set keeping constant the ratio between survived and dead patients. In order to limit the effect of class imbalance, we applied Synthetic Minority Oversampling Technique (*SMOTE*) [33] to the training set. Features were rescaled before feeding them to the classifier by removing the median and dividing by the interquartile range, as estimated on the training set [37], and five machine learning classifiers were used to develop patient survival prediction models. We considered the following classifiers: *Logistic Regression* (*LR*), *Support Vector Machine* (*SVM*) [34], *Decision Tree* (*Tree*), *Random Forest* (*RF*) [35] and *XGBoost* (*XGB*) [36] (evelution metric: *logloss* and objective function: *binary/logistic*). To evaluate the performance of the classifiers, Matthews correlation coefficients (*MCC*) [38] on the cross-validated test sets

**Table 3. Values, counts and percentages for each categorical variable of the dataset, stratified by *Survival* and for the full non-sepsis *SIRS* cohort.** DIAG.: Diagnosis, S: Survived, NS: Not Survived, EC: Elective, AC: Emergency, and M: Medical. E: Male, and K: Female.

| | | *SIRS* | | *SEPSIS* | |
|---|---|---|---|---|---|
| feature | value | counts | % | counts | % |
| DIAG. | S-AC | 32 | 3.92 | 10 | 2.26 |
| DIAG. | NS-AC | 2 | 0.25 | 1 | 0.23 |
| DIAG. | S-EC | 537 | 65.81 | 75 | 16.97 |
| DIAG. | NS-EC | 3 | 0.37 | 0 | 0 |
| DIAG. | S-M | 220 | 26.96 | 251 | 56.79 |
| DIAG. | NS-M | 20 | 2.45 | 103 | 23.30 |
| SEX | S-E | 464 | 56.86 | 209 | 47.29 |
| SEX | NS-E | 17 | 2.08 | 53 | 11.99 |
| SEX | S-K | 325 | 39.83 | 127 | 28.73 |
| SEX | NS-K | 8 | 0.98 | 51 | 11.54 |
| TOTAL | - | 814/816 | 99.75 | 440/440 | 100 |
| Survived | - | 789 | 96.93 | 336 | 76.36 |
| Not Survived | - | 25 | 3.07 | 104 | 23.64 |

were considered because its proven ability to summarize results from contingency tables and its invariance to class swapping [39–41, 60]. Specifically, the *MCC* can take values ranging from –1 to +1, where –1 represents the misclassification of all observations, 0 represents the random association, and 1 perfect classification. Average Receiver Operating Characteristic (ROC) curves and Precision-Recall curves (PRC) are also used to quantitatively assess the average model performances. Further details are reported in the supplementary material in Text A of S1 Appendix where additional sensitivity analyses summarizing model calibration on the test set (Text D of S1 Appendix) and the model performance with hyperparameters optimization (Text F of S1 Appendix) are also reported.

## Feature importance

The best-performing model for each cohort was selected and feature importance was estimated through single feature elimination (*SFE*) approach, that is by evaluating the MCC obtained

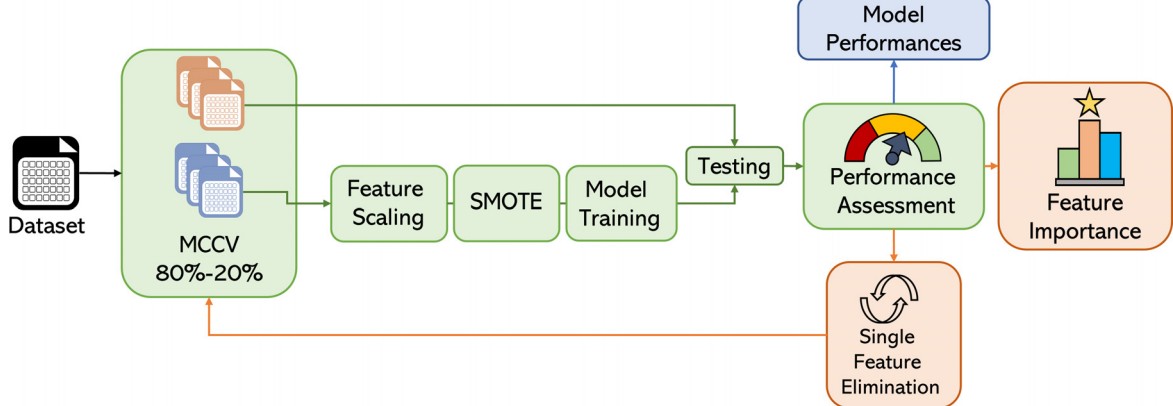

**Fig 1. Schematized representation of the proposed data processing flow.** MCCV: Monte Carlo Cross-Validation; SMOTE: Synthetic Minority Oversampling Technique.

after removing one variable at a time. In this case, the smaller the resulting MCC, the higher the importance of the variable which generated that observed drop in performances. Feature importance analysis was executed for 100 runs of Monte Carlo stratified cross-validation partitions with 80%/20% train-test split [42]. The resulting MCCs for each run are obtained from the test set observations. Finally, we used the Spearman correlation coefficient and the Kendall coefficient [43] to quantify the correlation between the obtained ranks. Both coefficients range from –1 to 1 (from anticorrelation to perfect matching) whereas the absence of correlation is given by a 0 coefficient.

## Results

### Associations between features and survival

Results of the statistical analysis are reported in Table 4. It can be observed that *APACHE II* and *SOFA* scores showed significant ($p<0.0001$) association with survival in both *SIRS* and *SEPSIS* cohorts. *EoC* resulted significantly associated ($p<0.0001$) with survival in *SIRS* cohort only.

### Survival prediction

Survival prediction performances for *SEPSIS* and *SIRS* cohorts are graphically summarized in Fig 2. Median MCCs, accuracy, sensitivity, specificity, F1-scores, positive predictive value, negative predictive value, areas under precision-recall and receiver operating characteristic curves, and the respective interquartile ranges are reported in the supplementary material (Text C of S1 Appendix). *SVM* and *LR* obtained the highest MCCs in predicting sepsis patient survival, that is 0.533 and 0.533, respectively. *Random Forest* performed as second best model in this cohort with MCC equal to 0.516 whereas *XGBoost* and *Tree* obtained the lowest results 0.459 and 0.368, respectively. *LR* was chosen as the best performing because of the highest third quartile.

The best score when predicting survival on the *SIRS* cohort was achieved with the *XGB* method that reached 0.489. The second and third best-performing models in the SIRS cohort were *RF* with MCC equal to 0.39 and *LR* showing MCC equal to 0.379. *SVM* and *Tree* showed scores equal to 0.378 and 0.289, respectively.

**Table 4. *p*-values obtained from the statistical analysis when testing associations with patient survival in the *SEPSIS* and *SIRS* cohorts.** Differences in numeric features between survived and deceased groups in the two cohorts were tested with the Mann-Whitney *U* test [44], whereas differences in categorical features were assessed with the $\chi^2$-test.

|  | **SIRS** | **SEPSIS** |
|---|---|---|
| Age | 0.0345 | 0.1460 |
| APACHE II | **<0.0001** | **<0.0001** |
| SOFA | **<0.0001** | **<0.0001** |
| CRP | 0.8693 | 0.0651 |
| WBCC | 0.1705 | 0.2607 |
| NeuC | 0.3202 | 0.4994 |
| LymC | 0.0255 | 0.9010 |
| EoC | **<0.0001** | 0.5651 |
| NLCR | 0.5184 | 0.8086 |
| PLTC | 0.4588 | 0.0744 |
| MPV | 0.2796 | 0.6240 |
| Sex | 0.4754 | 0.0540 |

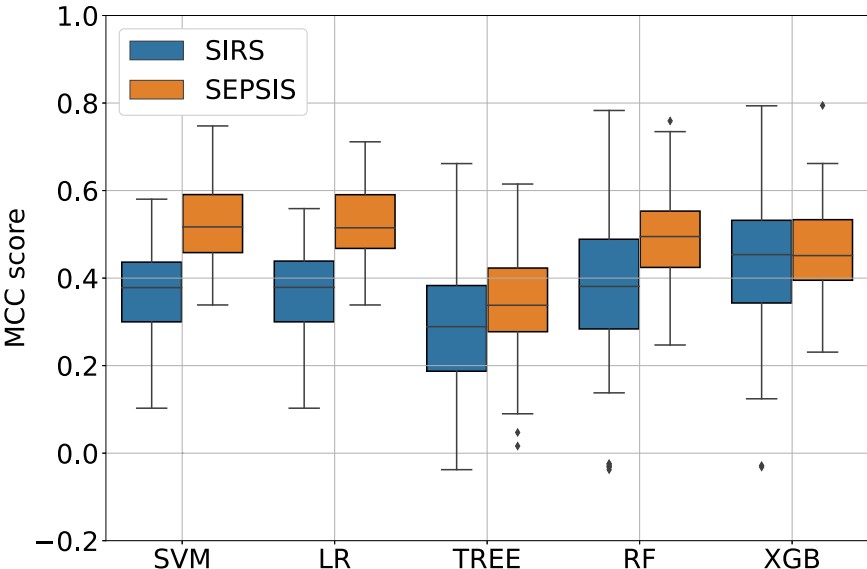

**Fig 2. Matthews correlation coefficients (*MCC*) of the different families of machine learning models in predicting *Survival* for the *SIRS* and *SEPSIS* cohorts.** Each violin plot shows the distribution of the data, whereas the small boxplot inside each violin plot shows the median and the first and third quartiles, the whiskers indicate 0.05 and 0.95 quantiles.

Fig 3 shows the median ROC and PRC curves for *SIRS* and *SEPSIS* cohorts as an overall summary of models' performances across all Monte Carlo runs.

## Feature ranking

This section describes the results obtained after the *SFE* approach performed on the models with the highest performance in the prediction task on each of the two cohorts. Median values and interquartile ranges for the resulting MCCs are reported in the supplementary material (Text E of S1 Appendix). A graphical representation of feature importance is shown in Fig 4a for the *SEPSIS* cohort and in Fig 4b for the *SIRS* cohort where features were ordered from lowest to the highest importance.

Specifically, APACHE II showed the highest importance, that is the lowest resulting median MCC equal to 0.436 (–0.097), in predicting *SEPSIS* patient survival with a *Logistic Regression* model. *MPV* ranked second in terms of feature importance for this specific cohort with MCC equal to 0.484. The other features did not induce a notable decrease in the model's performance. Feature ranking for the survival prediction of *SIRS* patients with *XGB* algorithm showed that *SOFA* has the highest importance with resulting MCCs equal to 0.381 (–0.108) when the feature is removed.

Results with Spearman coefficient and Kendall distance did not show a significant correlation between the two series, with correlation equal to –0.091 ($p = 0.737$) and –0.007 ($p = 0.983$), respectively.

## Discussion

Gucyetmez et al. [10] collected the data used in this study for exploring the ability of hemogram and *CRP* in discriminating between *SIRS* and *SEPSIS* cohorts. However, the authors did not investigate the prognostic role of the selected features within each cohort, therefore, our study aimed to investigate more in detail the importance of these features and the possible

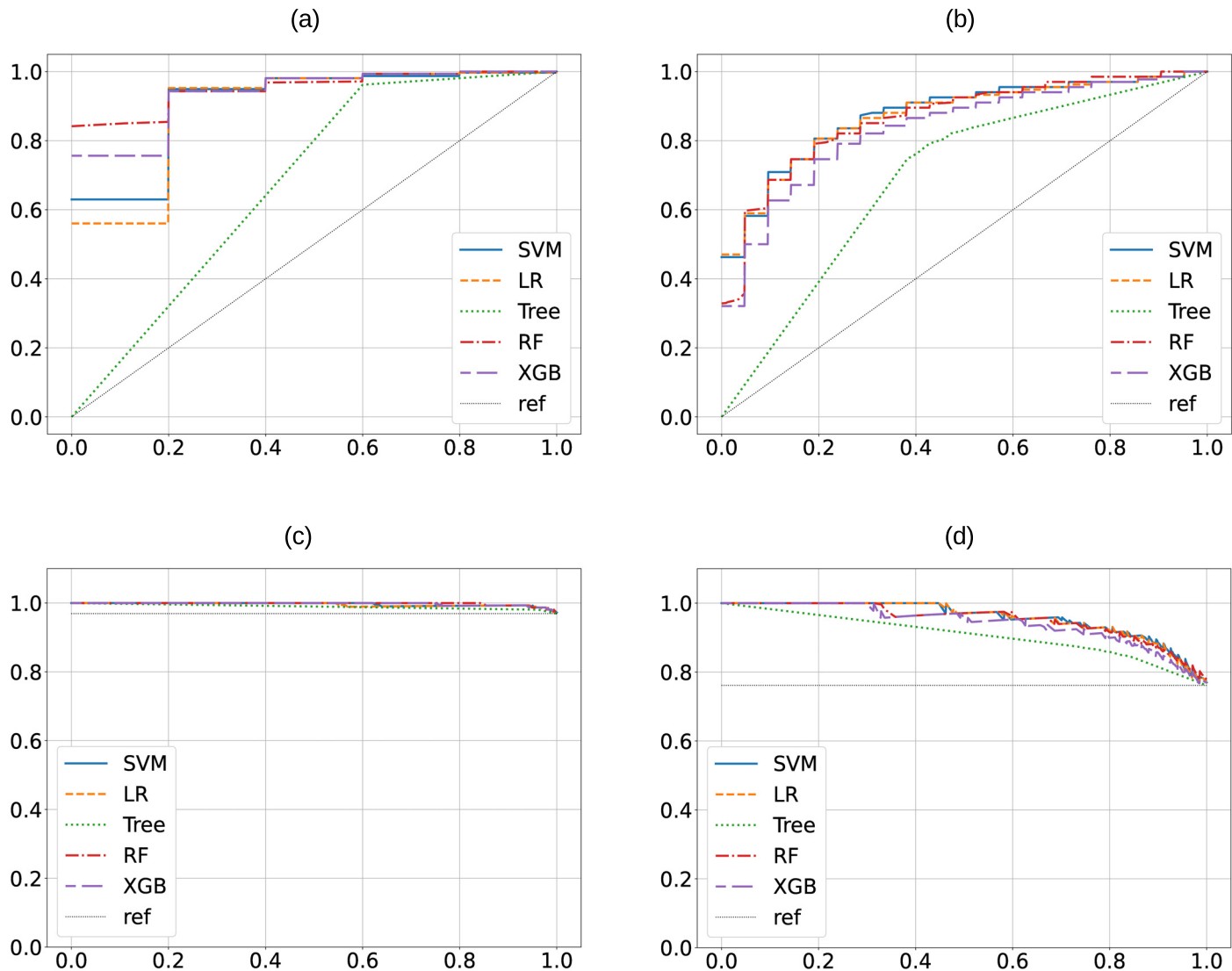

**Fig 3.** Receiver Operating Characteristic (ROC) curves for *SIRS* (panel (a)) and *SEPSIS* (panel (b)) cohorts showing the median performances of each model on the test sets generated during Monte Carlo cross-validation. Panels (c) and (d) depict Precision-Recall Curves (PRC) for *SIRS* and *SEPSIS* cohorts, respectively, showing the median performances of each model on the test sets generated during Monte Carlo cross-validation.

differences between the considered cohorts. Specifically, we performed the evaluation of the ability of hemogram parameters in predicting the survival of ICU patients diagnosed with *SIRS* or *SEPSIS*, using a set of parameters usually available in the patient clinical records. We used widely available features like patient sex, illness severity scores commonly measured and recorded at admission in the ICU, C-reactive protein, and blood cell count measurements. Patients' comorbidities were not available in the patient's records shared by Gucyetmez et al. despite they are commonly available in an ICU setting, which represents a significant lack of information. The developed models would have certainly benefited from more information about the patient's history and this could have led to a more precise identification of differences in the prognostic factors. Therefore, future studies will focus on the extension of these analyses on more complete data including patients' comorbidities.

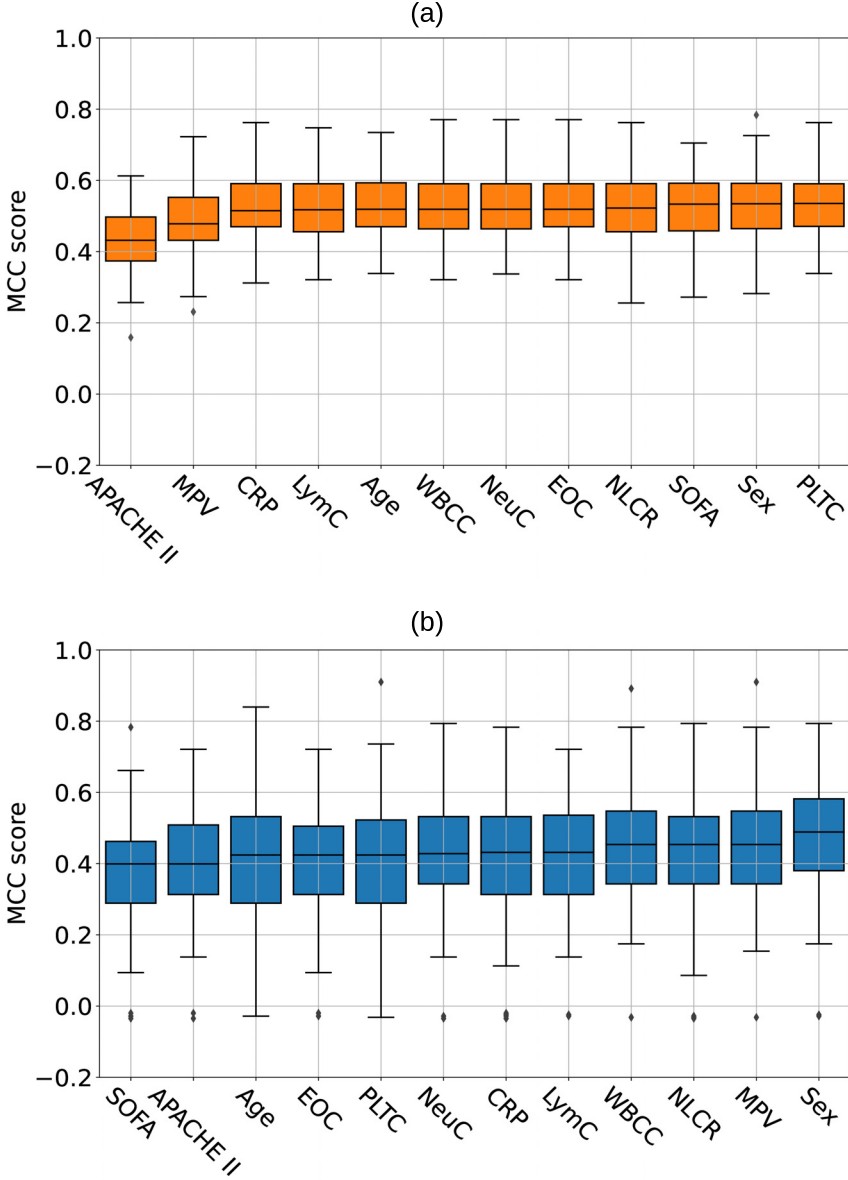

**Fig 4. Matthews correlation coefficients after single feature elimination for the *Survival* prediction task performed with (a) the *Logistic Regression* model in the *SEPSIS* cohort and (b) with the *XGBoost* model in the *SIRS* cohort.** Features were ranked according to importance from left to right. Each violin plot shows the distribution of the data, whereas the small boxplot inside each violin plot shows the median and the first and third quartiles, the whiskers indicate 0.05 and 0.95 quantiles.

The survival prediction models were developed and tested on *SIRS* and *SEPSIS* cohorts, with better performances observed in *SEPSIS* cohort.

Specifically, among all trained ML models, linear-based models like *LR* and *SVM* showed higher performances in the *SEPSIS* cohort, whereas *RF* and *XGB* performed better on the *SIRS* cohort.

Although average calibration curves are sub-optimal, which is likely due to the reduced sample size, the best-performing models show improved calibration with respect to the worst ones as expected.

This behavior suggests that a bigger population would allow for a proper calibration adjustment and translation of the model's output score to an even more precise individualized patient survival probability. Also, this approach might account for a possible covariate shift due to changes in patient characteristics, without the need for the development of a new model. Therefore, we do consider that the relationships between the available variables both intra- and inter-population can be considered a reliable multivariable comparison of the major factor predicting survival for both SIRS and sepsis patients. As it can be observed in Text F of S1 Appendix, the sensitivity analysis implementing the hyperparameter optimization shows results very close to those observed without hyperparameter optimization, thus highlighting the robustness of the proposed framework.

The feature importance analysis attributed the highest importance to *APACHE II* and *SOFA* scores for *SEPSIS* and *SIRS* cohorts, respectively, thus confirming the importance of a preliminary assessment of patients' risk at the admission in the ICU [45]. This result is also confirmed by statistical analysis, as shown in Table 4.

## *SEPSIS* cohort

Results on the *SEPSIS* cohort showed that *MPV* was the second most important variable in predicting survival. This result is in line with the observed association of a higher *MPV* with an increased mortality risk as well as its predictive role [46–48]. Our analysis ranked *EoC* and *CRP* as third and fourth most important features. In the literature, a lower *EoC* has been associated with mortality in critically ill medical patients [49], in patients admitted with an exacerbation of chronic obstructive pulmonary disease [50] and in patients with pneumonia [51]. Interestingly, although non-significant, our cohort showed an increase in *EoC* in deceased *SEPSIS* patients. An epidemiological study [52] pointed out that eosinophilia is a predictor of all-cause mortality and that an increased number of peripheral blood eosinophils may reflect an increased inflammatory response, resulting in tissue injury, a condition that may reflect our cohort. *CRP* was the fourth most important variable in our ML model. Of note, *CRP* had already shown the potential of being a predictor of survival of ICU patients [53], and more in general a predictor of mortality in ML frameworks [54].

## *SIRS* cohort

Interestingly, the third most important variable in predicting survival for the *SIRS* cohort was platelet counts, with a smaller median value for the non-survived patients than for the survived group. Indeed, Vanderschueren et al. [55] found that Thrombocytopenia was associated with a higher risk of death in a septic cohort, in line with our results considering the definition of sepsis (sepsis-1) used in 2000 which only required two SIRS criteria. CRP ranked fourth in predicting patient survival with non-sepsis SIRS and similar considerations as for the *SEPSIS* cohort can be done, moreover, its importance in predicting survival of a non-sepsis SIRS cohort was already observed in animal studies [56]. The fifth and sixth-ranked features were lymphocytes and eosinophils. In literature, Lymphocytes counts were found to be associated with increased mortality risk in general ICU patients [57], heart failure [58], and COVID-19 patients [59]. Eosinophils count significantly differed between survived and deceased groups with an increase in the deceased one. Similar considerations as for the *SEPSIS* cohort can be done also for eosinophil counts, where we already pointed out that this apparently opposite behavior with respect to literature might be due to the specific cohort of our study with undergoing inflammatory response [52].

### General considerations and applicability

The developed models show the ability to predict patient survival and specifically, this study can be considered as an important integration of the study performed by Gucyetmez et al. [10] so that once a patient with inflammatory response has been identified as septic or not the corresponding model can give us the possibility to immediately assess the likelihood of survival. Also, the feature importance analysis proposed in our study gives a clue on the main features that contributed to the developed cohort-specific score, and it suggests to clinicians which of the considered variables is more informative for a patient falling in the *SIRS* or *SEPSIS* cohort. It is important to notice that the SFE method presents some limitations when features are highly interdependent, since the contribution of a feature that is very important may still be underestimated due to the effect of other covariates that depend on it.

Finally, it is worth mentioning that we are not aware of whether these data were collected for administrative health reasons or whether they are commonly used for clinical practice, which might limit the general applicability of the results. However, the employment of data like these for scientific analyses based on computational intelligence can allow new scientific discoveries that otherwise would be impossible with traditional hospital technologies.

This study presents an original application of a statistical framework aimed at predicting patient survival. As the approach is mainly limited by the reduced sample size of the cohort, it is expected that a larger collection of data would allow for a more effective model calibration and optimization that would further improve the model generalizability, thus providing a more precise estimate of patient survival probability.

### Conclusions

The proposed study applies an original machine learning paradigm for processing clinical information at admission in the ICU to predict patient survival. The proposed approach relies on a multi-variable predictive modeling approach based on information gathered at the ICU admission, and aimed at predicting the likelihood of patient survival for patients with *SIRS* and with *SEPSIS*. Results provide insights into the differences of the most relevant variables between the two groups. A Monte Carlo Cross-Validation procedure was further applied to have robust estimates of the obtained scores. The performed sensitivity analysis showed that results did not notably vary with hyperparameter tuning thus confirming the need for a larger cohort to advance to a fully calibrated deployable model.

In this context, *Logisitic Regression* and *XGBoost* algorithms are the best-performing models for *SEPSIS* and *SIRS* cohorts, respectively. Moreover, feature importance analysis revealed a high importance of *APACHE II* score and a comparable important role of C-reactive protein in both cohorts. Also, *MPV* and *EoC* were revealed to be important predictors of survival mainly in the *SEPSIS* cohort, whereas they showed a secondary role in the *SIRS* cohort.

*SIRS* cohort showed greater importance of *SOFA* and platelets count features which instead ranked last in *SEPSIS*.

Importantly, beyond Gucyetmez et al. [10] findings, the proposed framework addresses the question of whether a patient has sepsis or not, and our models give clinicians the possibility to estimate patient's survival, as well as to identify the most important features involved in the stratification of patients' risk with *SIRS* or *SEPSIS*, and that also led to the proposed survival estimates.

Of note, to our knowledge, this is the first study where the ability of hemogram parameters in predicting patient survival at the admission in the ICU and the role of the considered features are investigated to highlight differences between *SIRS* and *SEPSIS* patients.

## Supporting information

**S1 Appendix.** Text A: Formulas of the confusion matrix rates. Text B: Model Hyperparameters. Text C: Model performances. Text D: Model Calibration. Text E: Single Feature Elimination. Text F: Sensitivity Analysis: Hyperparameter Optimization.
(PDF)

**S1 Fig. Average calibration curves across the MCCV runs for the best (left column) and worst (right column) for SEPSIS (upper row) and SIRS (lower row).**
(EPS)

## Author Contributions

**Conceptualization:** Davide Chicco.

**Data curation:** Davide Chicco.

**Formal analysis:** Maximiliano Mollura.

**Investigation:** Maximiliano Mollura, Davide Chicco, Alessia Paglialonga, Riccardo Barbieri.

**Methodology:** Maximiliano Mollura, Davide Chicco, Alessia Paglialonga, Riccardo Barbieri.

**Project administration:** Davide Chicco, Riccardo Barbieri.

**Resources:** Maximiliano Mollura, Davide Chicco.

**Software:** Maximiliano Mollura.

**Supervision:** Davide Chicco, Alessia Paglialonga, Riccardo Barbieri.

**Writing – original draft:** Maximiliano Mollura, Davide Chicco, Alessia Paglialonga, Riccardo Barbieri.

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
