## [Decision Letter · Decision Letter 0]

25 Apr 2023

PDIG-D-23-00039

Identifying prognostic factors for survival in intensive care unit patients with SIRS or sepsis by machine learning analysis on electronic health records

PLOS Digital Health

Dear Dr. Chicco,

Thank you for submitting your manuscript to PLOS Digital Health. After careful consideration, we feel that it has merit but does not fully meet PLOS Digital Health's publication criteria as it currently stands. Therefore, we invite you to submit a revised version of the manuscript that addresses the points raised during the review process.

Please submit your revised manuscript within 60 days Jun 24 2023 11:59PM. If you will need more time than this to complete your revisions, please reply to this message or contact the journal office at digitalhealth@plos.org. Please include the following items when submitting your revised manuscript:

We look forward to receiving your revised manuscript.

Kind regards,

Nan Liu

Academic Editor

PLOS Digital Health

Journal Requirements:

1. We ask that a manuscript source file is provided at Revision. Please upload your manuscript file as a .doc, .docx, .rtf or .tex.

2. Please insert an Ethics Statement at the beginning of your Methods section, under a subheading 'Ethics Statement'. It must include:

1) The name(s) of the Institutional Review Board(s) or Ethics Committee(s)

2) The approval number(s), or a statement that approval was granted by the named board(s) 

3) (for human participants/donors) - A statement that formal consent was obtained (must state whether verbal/written) OR the reason consent was not obtained (e.g. anonymity). NOTE: If child participants, the statement must declare that formal consent was obtained from the parent/guardian.

3. Please provide separate figure files in .tif or .eps format only and remove any figures embedded in your manuscript file. Please also ensure that all files are under our size limit of 10MB.

4. We have noticed that you have uploaded Supporting Information files, but you have not included a list of legends. Please add a full list of legends for your Supporting Information files after the references list.

Additional Editor Comments (if provided):

Thank you for submitting this study. While the manuscript has merits, some clarifications such as methodology and clinical utility are needed, as suggested by the reviewers.

Reviewers' comments:

Reviewer's Responses to Questions

**Comments to the Author**

1. Does this manuscript meet PLOS Digital Health’s publication criteria? Is the manuscript technically sound, and do the data support the conclusions? The manuscript must describe methodologically and ethically rigorous research with conclusions that are appropriately drawn based on the data presented.

Reviewer #1: Yes

Reviewer #2: Partly

2. Has the statistical analysis been performed appropriately and rigorously?

Reviewer #1: No

Reviewer #2: No

3. Have the authors made all data underlying the findings in their manuscript fully available (please refer to the Data Availability Statement at the start of the manuscript PDF file)?

Reviewer #1: Yes

Reviewer #2: No

4. Is the manuscript presented in an intelligible fashion and written in standard English?

Reviewer #1: Yes

Reviewer #2: Yes

5. Review Comments to the Author

Reviewer #1: The study by Mollura et. al., applied supervised machine-learning models to data collected from patients admitted to intensive care unit to highlight differences in clinical features for survival. 

There are a number of major issues that need to be addressed:

Abstract:

1. "However, major differences between these two populations when estimating patients’ risk after ICU admission are not fully elucidated." 

Which subsequent risk is being referred to in this sentence?

Author summary:

2. "However, distinguishing between a systemic inflammatory response syndrome (SIRS) and sepsis is not straightforward and the major differences related with patients’ risk are not fully elucidated."

The authors indicate the inability to differentiate between SIRS and sepsis as the major concern.

And this point is reiterated in the introduction.

Why the focus on prognostic factors for survival after either SIRS or sepsis?

There is some disconnect.

Introduction:

3. Why the distinction between patients with SIRS and sepsis?

Will it make a difference if the authors just identified the cohort as patients admitted to ICU, in general?

The aim of the study is to predict survival in ICU patients.

Methods:

4. The dataset did not have information on comorbid conditions at the time of ICU admission. The presence of comorbid conditions impacts on survival of ICU patients. This was not accounted in the models. Does this impact on the models?

5. "It is important to notice that these data were collected originally for administrative health reasons, and not for scientific purpose." The authors considered this a strength and did not highlight potential limitations of using such administrative data for a scientific purpose.

6. Missing data and how missing data was handled was not discussed in the manuscript.

7. Association between features and survival - how was this assessed? The sentence does not provide the details: "Association between the input features and patient survival was also explored with classical statistical approaches."

8. A study flow diagram will be needed to help explain what was done.

9. Was the data split for the 2 analyses: SIRS-related analysis using data from 816 patients and sepsis-related analysis using data from 441 patients?

10. Statistically, was the study adequately powered?

11. Can the authors clarify which model is "Decision Tree (Tree)"? Random forest and XGBoost are decision tree-based algorithms or models.

12. Could the authors explain why this was done and provide reference for this: "Features were standardized removing the median and dividing by the interquartile range, both estimated on the training set, and five machine learning classifiers were used to develop patient survival prediction models."

13. Is Monte Carlo stratified cross-validation a cross-validation approach or a feature importance or ranking approach?

14. Will it be helpful if area-under the curve and calibration plots are provided to assess the performance of the models in addition to MCC?

15. Page 6 has Fig 2 showing ROC curves - there is no mention of ROC curve in the methods section. 

Minor concern:

1. The authors use the word "multivariate" a number of times in the manuscript. This should be "multivariable".

Reviewer #2: Mollura et al present in this paper a machine learning approach to predict survival of patients admitted to the ICU having systemic inflammatory response syndrome or sepsis. The authors investigate the performance of five machine learning models and subsequently estimate the importance of the input features. The paper is well written and states clearly the intentions. However, I have some concerns in relation to how the results are generated methodologically and therefore the overall value of the work is somewhat unclear. I also have some doubts on the actual clinical relevance of the paper. I acknowledge that a prototype of a method can be valuable, even if it in its present form would not be finalized for implementation in an ICU electronic patient record system. 

Major concerns:

I really appreciate that the authors use a correlation coefficient for evaluating model performance – and in this way take into account all elements of the confusion matrix. Nevertheless, I find it somewhat misleading that the outcome encoded as 1 is survival and not death (table 1 says that the feature ‘mortality’ encodes death as 1, correct me if this means that you encoded death in the outcome with 1 - in that case you can ignore the rest of this point). Using survival as target outcome is counterintuitive, since the rare event usually is the one encoded in this way. To this end I believe that the design makes AUROC, MCC and NPV informative metrics only. Performances in the supplementary tables are exceptionally good, with AUPRC values of > 95% in most cases. An AUPRC of 0.998% implies you are extremely good at predicting the positive class, but in this case the positive class is survival, which accounts for the majority of the population (~97% for the SIRS population). This would have not been the case if death was encoded as 1. Even if this is not an error, I believe that this decision makes the quantitative interpretation of the metrics misleading. Another note is that for the metrics that are not “areas under …” it is not mentioned explicitly what the probability threshold used for the calculation is. Calibration curves are not included at all, which makes the clinical utility of the model less clear. 

It is not stated at what timepoint the features were included and at what timepoint the survival was assessed. The authors also do not explain how multiple measurements were aggregated. This is very important in order to understand the study design. Also, no information about imputation and missing values has been included. 

The authors did not provide any information about which hyperparameters were used to run the five different ML models. This is also a very important aspect of the study design.

I do not fully understand the way the feature importance was calculated. RFE should routinely remove the least important feature (evaluated using MCC, according to what the authors state), until a desired number of features is reached. In Results, the MCC is reported for each model with only one feature knocked out at a time. This makes the whole feature interpretation difficult to assess as most of the features are not independent or even included into one another (e.g. age into APACHE). This causes features like age, which is usually a very strong predictor of ICU survival, not important because even if explicitly removed from the feature set, it is still implicitly used for prediction by including the APACHE feature. The same argument is also valid for other features. 

It is unclear whether multiple test correction has been used to evaluate the associations between features and survival ?

The code used for the analysis has not been provided. This is a very important aspect. 

Minor concerns:

I can see that LOS-ICU and diagnosis have not been included in the feature set. While I understand the reason this has been done for LOS-ICU (leakage of information as this feature is only available after the outcome actually happens), I do not fully understand why ‘diagnosis’ has not been used. 

The statement on normalization is mentioned twice: “Features were standardized removing the median and dividing by the interquartile range, …” and immediately after “Features were rescaled before feeding them to the classifier by removing the median and dividing by the interquartile range …”. 

The feature ‘sex’ is indicated as gender in the data set section, please standardize. 

In my opinion, this sentence does not really belong to the dataset section : “In this retrospective study, we developed predictive models of patient survival using machine learning algorithms and we evaluated the importance of features associated with patient survival using machine learning and biostatistics approaches. “

The authors used violin plots. These plots have the disadvantage of showing the full tail of a distribution. In Figure 1 for example many models have MCC scores lower than 0 (it is a bit surprising to have such a wide range of MCC values) . I would suggest instead to use a boxplot, to limit the visualization to the interquartile range and outliers. 

Conclusion: while the paper has many valuable aspects and is well written, the unclear methodology and the lack of clear technical descriptions make it very hard to assess the value of the approach.

6. PLOS authors have the option to publish the peer review history of their article (what does this mean?). If published, this will include your full peer review and any attached files.

**Do you want your identity to be public for this peer review?** For information about this choice, including consent withdrawal, please see our Privacy Policy.

Reviewer #1: No

Reviewer #2: No

---

## [Decision Letter · Decision Letter 1]

31 Jul 2023

PDIG-D-23-00039R1

Identifying prognostic factors for survival in intensive care unit patients with SIRS or sepsis by machine learning analysis on electronic health records

PLOS Digital Health

Dear Dr. Chicco,

Thank you for submitting your manuscript to PLOS Digital Health. After careful consideration, we feel that it has merit but does not fully meet PLOS Digital Health's publication criteria as it currently stands. Therefore, we invite you to submit a revised version of the manuscript that addresses the points raised during the review process.

Please submit your revised manuscript within 60 days Sep 29 2023 11:59PM. If you will need more time than this to complete your revisions, please reply to this message or contact the journal office at digitalhealth@plos.org. Please include the following items when submitting your revised manuscript:

We look forward to receiving your revised manuscript.

Kind regards,

Nan Liu

Academic Editor

PLOS Digital Health

Journal Requirements:

Additional Editor Comments (if provided):

Thank you for the revision. While we see some parts of the manuscript have been improved, there are still major concerns, such as calibration and clinical relevance, among others. Please carefully address reviewers' concerns and comments, provide detailed responses, and make necessary further revisions. It is important to ensure your research results are translatable into clinical practice.

Reviewers' comments:

Reviewer's Responses to Questions

**Comments to the Author**

1. If the authors have adequately addressed your comments raised in a previous round of review and you feel that this manuscript is now acceptable for publication, you may indicate that here to bypass the “Comments to the Author” section, enter your conflict of interest statement in the “Confidential to Editor” section, and submit your "Accept" recommendation.

Reviewer #1: (No Response)

Reviewer #2: (No Response)

2. Does this manuscript meet PLOS Digital Health’s publication criteria? Is the manuscript technically sound, and do the data support the conclusions? The manuscript must describe methodologically and ethically rigorous research with conclusions that are appropriately drawn based on the data presented.

Reviewer #1: Partly

Reviewer #2: Partly

3. Has the statistical analysis been performed appropriately and rigorously?

Reviewer #1: No

Reviewer #2: No

4. Have the authors made all data underlying the findings in their manuscript fully available (please refer to the Data Availability Statement at the start of the manuscript PDF file)?

Reviewer #1: Yes

Reviewer #2: No

5. Is the manuscript presented in an intelligible fashion and written in standard English?

Reviewer #1: Yes

Reviewer #2: Yes

6. Review Comments to the Author

Reviewer #1: (No Response)

Reviewer #2: The authors have in technical terms replied to all the points, however, calibration for example is about making a method work in practice or at least substantiating that it has that potential. The are also other points I believe are not fully clear and overall the paper is not ready for publication in my view. 

My main concerns are in particular the two aspects, model calibration and feature interpretation. I understand and agree that calibration curves in cases where the dataset is very imbalanced are often far from the actually observed probabilities. But in these cases one would expect to see mitigation with Platt scaling or isotonic regression on the probabilities. If the decision is to proceed without calibration adjustment, I would still include those calibration curves in the supplementary and specify in the Discussion this as a limitation. The model as it stands can only be used to classify two groups depending on a single decision threshold and cannot provide an individualized survival probability – and this is in my view a limiting factor. Another comment related to the revised version is that the authors did not use any hyperparameter search. As they have shown in the supplementary materials, they used the models with the default parameters as directly provided by scikit-learn. This may not be a problem per-se but I wonder if they could have obtained better performances if they had explored other model configurations. This again makes this translational value unclear. It would even be unethical to use a method in the clinic that was suboptimal, given the data, and we do not know if this is the case. 

The feature interpretation is still not clear to me. The response to my previous question “In particular, we use Recursive Feature Elimination, i.e. we remove iteratively only one feature at a time to analyze the observed loss in prediction performances after training and testing again the algorithm. ” seems not to be correct. RFE does not iteratively remove only one feature at a time. Reference 42 used to explain the method says .. “(a) running RF to determine initial importance scores, (b) removing the bottom 3% of variables with the lowest importance scores from the data set (3% was chosen because of the high computational demands of using a lower threshold; this resulted in a total of 324 RF runs), and (c) assigning ranks to removed variables according to the order in which they were removed and their most recent importance scores (i.e., importance scores are only compared within runs, not between runs). This was performed iteratively using the reduced data set created in step two until 3% of the "number of remaining variables" rounds to zero (i.e., no further variables could be removed).” This definition is also consistent with how sklearn (which was not used by the authors to calculate RFE in their notebook) defines it (https://scikit-learn.org/stable/modules/generated/sklearn.feature_selection.RFE.html). The method "removes one feature at a time" to my knowledge would not account for correlated variables as it is done in reference 42 with RFE. 

This paper does not really propose a novel framework (as the authors state in the Discussion), and overall I think the implementation of the standard methods along with the limited clinical utility of it makes the paper somewhat incremental.

7. PLOS authors have the option to publish the peer review history of their article (what does this mean?). If published, this will include your full peer review and any attached files.

**Do you want your identity to be public for this peer review?** For information about this choice, including consent withdrawal, please see our Privacy Policy. 

Reviewer #1: No

Reviewer #2: No

---

## [Decision Letter · Decision Letter 2]

26 Oct 2023

PDIG-D-23-00039R2

Identifying prognostic factors for survival in intensive care unit patients with SIRS or sepsis by machine learning analysis on electronic health records

PLOS Digital Health

Dear Dr. Chicco,

Thank you for submitting your manuscript to PLOS Digital Health. After careful consideration, we feel that it has merit but does not fully meet PLOS Digital Health's publication criteria as it currently stands. Therefore, we invite you to submit a revised version of the manuscript that addresses the points raised during the review process.

Please submit your revised manuscript within 60 days Dec 25 2023 11:59PM. If you will need more time than this to complete your revisions, please reply to this message or contact the journal office at digitalhealth@plos.org. Please include the following items when submitting your revised manuscript:

We look forward to receiving your revised manuscript.

Kind regards,

Nan Liu

Academic Editor

PLOS Digital Health

Journal Requirements:

Additional Editor Comments (if provided):

Thank you for addressing Reviewer 2's comments. We noticed that Reviewer 1's comments were accidentally stored in the wrong place, thus you were unable to see them. Below please find Reviewer 1's comments for Revision 1 & Revision 2. Apologies for any inconvenience.

Reviewer 1 comments on Revision 1

Thank you to Mollura and colleagues for the effort in addressing the initial questions and suggestions - very much appreciated.

I am, however, unsure about the clinical relevance and how this study could possible possible impact patient care. My main concerns are:

1. From the response, the data being used for this study is from the study: https://doi.org/10.1371/journal.pone.0148699. This data was collected over a decade ago. Patient characteristics and disease profiles change based on advances in medical practice. Over the past decade, there has been significant advances and patient profiles have changed. What is the impact of this on the study findings? Risk predictions models, for example, are revised/updated to deal which such changes.

2. The research idea and concept is great, but is the data being used the most appropriate to address this research question?

3. Calibration curves are highlighting significant imbalanced data. The models are consistently underestimating real probabilities. Could this be discussed/highlighted in the discussion section? How will this possibly impact on clinical implications?

Reviewer 1 comments on Revision 2

My review was not made available to the authors in the decision letter, hence these have not been addressed.

Additionally, based on authors' response to concerns from the other reviewer, I am not convinced about the clinical importance and how the findings can be translational at this stage.

Major concerns:

1. Small sample size

2. No external validation of the models that have been developed. The current model after further sensitivity analysis is overfitting.

3. Authors choosing to change labels of methods supposedly used in the analysis without evidence to support the label.

Many thanks for the opportunity to review this.

Kind regards

Reviewers' comments:

Reviewer's Responses to Questions

**Comments to the Author**

1. If the authors have adequately addressed your comments raised in a previous round of review and you feel that this manuscript is now acceptable for publication, you may indicate that here to bypass the “Comments to the Author” section, enter your conflict of interest statement in the “Confidential to Editor” section, and submit your "Accept" recommendation.

Reviewer #1: (No Response)

Reviewer #2: All comments have been addressed

2. Does this manuscript meet PLOS Digital Health’s publication criteria? Is the manuscript technically sound, and do the data support the conclusions? The manuscript must describe methodologically and ethically rigorous research with conclusions that are appropriately drawn based on the data presented.

Reviewer #1: (No Response)

Reviewer #2: Yes

3. Has the statistical analysis been performed appropriately and rigorously?

Reviewer #1: No

Reviewer #2: Yes

4. Have the authors made all data underlying the findings in their manuscript fully available (please refer to the Data Availability Statement at the start of the manuscript PDF file)?

Reviewer #1: Yes

Reviewer #2: Yes

5. Is the manuscript presented in an intelligible fashion and written in standard English?

Reviewer #1: Yes

Reviewer #2: Yes

6. Review Comments to the Author

Reviewer #1: (No Response)

Reviewer #2: Unfortunately some sections were modified but not marked in blue. Nevertheless I think the authors addressed all the comments I had. I would add in limitations that the SFE method is not fully informative when features are highly interdependent – since the contribution of a feature that is very important may still result as not important during SFE because of its contribution through other covariates that depend on it. With this additional remark in the Discussion I believe the manuscript is ready for publication.

7. PLOS authors have the option to publish the peer review history of their article (what does this mean?). If published, this will include your full peer review and any attached files.

**Do you want your identity to be public for this peer review?** For information about this choice, including consent withdrawal, please see our Privacy Policy. 

Reviewer #1: No

Reviewer #2: No

---

## [Editor Report · Decision Letter 3]

5 Feb 2024

Identifying prognostic factors for survival in intensive care unit patients with SIRS or sepsis by machine learning analysis on electronic health records

PDIG-D-23-00039R3

Dear Dr. Chicco,

We are pleased to inform you that your manuscript 'Identifying prognostic factors for survival in intensive care unit patients with SIRS or sepsis by machine learning analysis on electronic health records' has been provisionally accepted for publication in PLOS Digital Health.

Best regards,

Nan Liu

Academic Editor

PLOS Digital Health